# Metabolic Features of Tumor Dormancy: Possible Therapeutic Strategies

**DOI:** 10.3390/cancers14030547

**Published:** 2022-01-21

**Authors:** Erica Pranzini, Giovanni Raugei, Maria Letizia Taddei

**Affiliations:** 1Department of Experimental and Clinical Biomedical Sciences “Mario Serio”, University of Florence, Viale Morgagni 50, 50134 Florence, Italy; erica.pranzini@unifi.it; 2Department of Experimental and Clinical Medicine, University of Florence, Viale Morgagni 50, 50134 Florence, Italy; marialetizia.taddei@unifi.it

**Keywords:** dormancy, metabolism, tumor recurrence, circulating tumor cells, cancer stem cells

## Abstract

**Simple Summary:**

Tumor recurrence still represents a major clinical challenge for cancer patients. Cancer cells may undergo a dormant state for long times before re-emerging. Both intracellular- and extracellular-driven pathways are involved in maintaining the dormant state and the subsequent awakening, with a mechanism that is still mostly unknown. In this scenario, cancer metabolism is emerging as a critical driver of tumor progression and dissemination and have gained increasing attention in cancer research. This review focuses on the metabolic adaptations characterizing the dormant phenotype and supporting tumor re-growth. Deciphering the metabolic adaptation sustaining tumor dormancy may pave the way for novel therapeutic approaches to prevent tumor recurrence based on combined metabolic drugs.

**Abstract:**

Tumor relapse represents one of the main obstacles to cancer treatment. Many patients experience cancer relapse even decades from the primary tumor eradication, developing more aggressive and metastatic disease. This phenomenon is associated with the emergence of dormant cancer cells, characterized by cell cycle arrest and largely insensitive to conventional anti-cancer therapies. These rare and elusive cells may regain proliferative abilities upon the induction of cell-intrinsic and extrinsic factors, thus fueling tumor re-growth and metastasis formation. The molecular mechanisms underlying the maintenance of resistant dormant cells and their awakening are intriguing but, currently, still largely unknown. However, increasing evidence recently underlined a strong dependency of cell cycle progression to metabolic adaptations of cancer cells. Even if dormant cells are frequently characterized by a general metabolic slowdown and an increased ability to cope with oxidative stress, different factors, such as extracellular matrix composition, stromal cells influence, and nutrient availability, may dictate specific changes in dormant cells, finally resulting in tumor relapse. The main topic of this review is deciphering the role of the metabolic pathways involved in tumor cells dormancy to provide new strategies for selectively targeting these cells to prevent fatal recurrence and maximize therapeutic benefit.

## 1. Tumor Dormancy: Much More Than a Simple Cell Cycle Arrest

### 1.1. Cellular Dormancy as an Adaptive Strategy for Tumor Progression

Tumor dormancy is defined as a temporary, reversible mitotic and growth arrest [1] physiologically activated by the organisms to adapt to stressful conditions. In oncology, tumor dormancy is a phase of cancer progression characterized by the presence of the disease, which remains in an undetectable phase before re-acquiring proliferative traits [2]. Tumor dormancy can be separated into two distinct phenomena: “tumor mass dormancy” and “cellular dormancy”. The first one refers to the arrest of cancer cell proliferation within the tumor mass caused by apoptosis, owing to poor vascularization and consequent hypoxia (“angiogenic dormancy”) [3,4] or by the immune response (“immune dormancy”) [5,6]. Under these conditions, cancer cells are not inactive, but the lesion does not expand due to cell death antagonizing cellular proliferation. Conversely, the term “cellular dormancy” refers to the process toward which cancer cells undergo a reversible G0 cell cycle arrest through the activation of quiescence programs [7]. Dormant cells are characterized by a reversible state during which they cease to divide but retain the ability to re-enter the cell cycle. Quiescence/dormancy originates from altered intrinsic signaling or the lack/alteration of contextual cues on which cancer cells previously depended [8]. The terms “dormancy” and “quiescence” are often used interchangeably, even if some authors have classified dormancy as having a more profound, persistent arrested state than quiescence [9]. Alongside, dormancy differs from senescence as it represents an irreversible state activated in normal cells in response to replicative or oncogenic stresses [10]. Although this definition is widely still in use, recent evidence suggests that senescence can also be reversible and may represent a causal link to disease recurrence [11].

Dormancy represents a basic survival strategy providing a selective advantage for organisms to persist in a hostile environment. Tumor cell dormancy, therefore, exploits evolutionarily conserved mechanisms of adaptation to succeed during stressful steps in tumor progression [12]. From this perspective, dormancy is a crucial element driving the cancer cells’ outcome in the patient’s organism, allowing them to survive physiological and therapy-induced insults. Specifically, tumor dormancy participates and supports crucial steps of cancer progression, such as primary tumor initiation (“local tumor dormancy”), metastatic dissemination (“metastatic dormancy”), and escape from anti-cancer therapies (“therapy-induced dormancy”) [8], thereby driving two of the major challenges in clinical oncology: metastatic diseases and cancer recurrence after therapy. Moreover, inside a tumor, a subpopulation of “dormancy-competent cancer stem cells” may alternate phases of dormancy and rapid growth [13,14], ensuring long-term tumor maintenance and supporting drug resistance, metastatic potential and ultimate disease recurrence [15].

In the early phases of tumorigenesis, dormant cells may accumulate mutations in a dynamic multistep process characterized by a hyper-proliferative reactivation followed by a period of dormancy from which cancer cells may exit to undergo a final malignant transformation, lastly resulting in tumor mass formation [16].

During the multistep process of metastasis formation, a small fraction of “disseminating tumor cells” (DTCs) invade surrounding tissues, intravasate, survive the circulation, extravasate, and finally colonize the secondary organ, where they need to adapt to the new non-permissive microenvironment and re-start proliferating, ultimately resulting in a second metastatic lesion [17]. Metastasis was traditionally described as a tardive event in the complex process of tumorigenesis, occurring only when cancer cells acquire the necessary genetic/epigenetic and metabolic adaptations to succeed all the steps of the metastatic process [18,19]. However, mounting evidence demonstrates that cancer cell dissemination may occur for the entire duration of tumor development [20], long before acquiring all the molecular signatures for metastasis formation. This implies that early DTCs require a period of adaptation and molecular evolution before a proliferating restart in the secondary organ site. The entry in a dormant phase allows DTCs to acquire molecular features for adapting to the new microenvironment and initiate colonization [7,21].

Besides its role in quiescent cells, in the primary tumor and the arising metastatic lesion, dormancy is also a key feature of cancer cells surviving the treatment with anti-proliferative chemotherapeutics drugs. These cells, also termed “drug-tolerant persister cells” (DTPCs), can be considered as a “minimal residual disease”, able to escape the selective pressure of anti-cancer drugs by entering into a dormant condition [8]. It is still not completely clear whether DTPCs are present in the cancer cell population before the treatment [22] or whether drug exposure actively induces a phenotypic transition to a dormant state in these cells [23]. Many molecular mechanisms have been involved in drug resistance, such as increased rates of drug efflux, DNA damage repair, cell death inhibition and metabolic changes that specifically promote drug inhibition and/or degradation, or precise mutation of drug targets [24,25]. Unlike drug resistance, therapy-induced dormancy is not characterized by the acquisition of specific genetic mutations of drug targets [24] but represents a transient non-mutational phenotype in which cancer cells survive the treatment as they do not proliferate. Following drug withdrawal, DTPCs generally resume proliferation, maintaining parental cells’ drug sensitivity [22,26]. However, in some cases, due to epigenetic reprogramming, DTPCs may also accumulate profound transcriptional alterations of resistance markers, leading to drug resistance [27]. In this view, “drug tolerance” could represent a stage between sensitivity and resistance during which the acquisition of a dormant phenotype represents a selective advantage to cancer cells to acquire resistance properties [28]. Indeed, a slowdown of the proliferation rate provides a selective advantage to cancer cells to survive drug pressure and acquire resistance features [29,30]. Moreover, dormant cells may also modify the environment, creating an immune-tolerant framework that further facilitates drug resistance [31,32].

In summary, the ability to undertake a period of dormancy represent a necessary adaptation for cancer cells to support metastatic dissemination and tumor relapse following therapy. Therefore, understanding the molecular basis and the metabolic strategies of cancer cells undergoing dormancy is essential for developing efficacious approaches to prevent tumor relapse.

### 1.2. Multiple Mechanisms Driving Tumor Cell Dormancy

Cancer cell dormancy prevents cell death under stressful conditions by inducing a G0 cell cycle arrest which results from both the alteration of intrinsic and autocrine signaling, and the occurrence of external-inducing inputs derived from the microenvironment [33].

#### 1.2.1. Intracellular Mechanisms

Several alterations in genes associated with proliferation and differentiation correlate with the initiation and maintenance of the dormant phenotype. For instance, the overexpression of the FBXW7 gene, encoding a component of specific E3 ubiquitin ligase, leading to the degradation of the positive cell-cycle regulators cyclin-E and c-Myc, is essential for the maintenance of dormancy in breast cancer DTCs [34]. Similarly, the overexpression of the KiSS1 gene is associated with dormancy in breast and ovarian cancers, as well as in melanoma [35,36,37].

Moreover, the phenotypic plasticity characterizing cellular dormancy is mostly governed by dynamic epigenetic mechanisms [14,38]. In ovarian cancer, the expression of the genes TIMP3 and CDH1 is raised during dormancy and is reduced during the transition to active proliferation by alterations in DNA methylation and histone modification [39]. Similarly, the epigenetic dioxygenase responsible for the oxidation of genomic 5-methylcytosine to 5-hydroxymethylcytosine TET2 is critical in cancer cells transiting to a slow-cycling state [40]. The nuclear receptor subfamily 2 group F member 1 (NR2F1), a key regulator of tumor dormancy, is epigenetically up-regulated by promoter hypermethylation in experimental head and neck squamous cell carcinoma (HNSCC) dormancy models and DTCs from prostate cancer patients carrying long-term dormant disease [41]. Moreover, non-coding RNAs are extensively involved in modulating dormancy–proliferation cycles. Remarkably, a consensus set of dormancy-associated miRs (DmiRs) has been identified as the principal regulation pattern correlating with the switch from dormancy to fast-growth in breast carcinoma, glioblastoma, and osteosarcoma [42]. Moreover, in metastatic breast cancer, gap junctions-mediated and exosomal transfer of miRNAs from the bone marrow (BM) stromal cells promote cancer cell dormancy in the BM niche [43,44].

Importantly, cancer cell dormancy can also be controlled by alterations in the main survival/cell cycle signaling pathways. In particular, the ERK/p38α balance plays a central role in modulating the dormant phenotype by inducing and integrating different signaling pathways in cancer cells [45]. Specifically, p38α signaling protects dormant cancer cells from stress by inducing the unfolded protein response (UPR) through the up-regulation of ER chaperone BiP and PERK activation [46]. Moreover, low ERK1/2, in concomitance with high p38 activation, induces a G0-G1 cell cycle arrest through the up-regulation of different transcriptional factors (including p53, NR2F1, and basic helix-loop-helix domain containing, class B, 3 (BHLHB3)) and the down-regulation of FOXM1 and c-Jun, which promote G1 exit [47,48,49]. Furthermore, under additional stresses, a low ERK/p38α ratio may also favor the expression of the chaperone BiP/Grp78, which inhibits Bax activation to avoid apoptosis and ensure adaptive survival to dormant cells [50].

Finally, the absence of proliferation-inducing stimuli may also lead to dormancy. In pancreatic cancer, the ablation of the oncogenic drivers mutant KRAS and c-MYC induces a dormant state, allowing the survival of a few residual cancer cells that rely on autocrine IGF1/AKT as an adaptive mechanism [51]. Pancreatic cells surviving KRAS oncogene ablation show dormancy, are responsible for tumor relapse, and mostly rely on oxidative phosphorylation (OXPHOS) for survival [52].

Together, the induction of these intrinsic pathways results in the activation of response to stress strategies such as autophagy [53] and the UPR [54] as two major adaptations in dormant cells. Irrespectively of the dormancy-inducing stimuli, autophagy sustains the acquisition of a dormant phenotype by providing appropriate energy balance under metabolic stress and by managing the reactive oxygen species (ROS) accumulation [55]. For example, the re-expression of the tumor suppressor Aplasia Ras homolog member 1 (ARHI; DIRAS3) in breast and ovarian cancer cells prevents apoptotic cell death and promotes dormancy by inducing autophagy [56,57,58] and altering fundamental metabolic pathways [59]. In pancreatic duct adenocarcinoma (PDAC), characterized by abnormal copper accumulation, targeting the copper transporter 1 (SLC31A1), or the treatment with a copper chelator, induce a dormancy-like response sustained by increased autophagy to resist cell death both in in vitro and in vivo models [60]. In osteosarcoma, dormancy protects cells against chemotherapeutics and preserves cell survival by favoring an autophagic state [61]. Coherently, dormancy correlates with the transcriptional up-regulation of driving autophagy-related genes, such as LC3, ATG4, ATG5, ATG7, and BECN1 [62,63]. The importance of autophagy in cancer dormancy was also demonstrated in ovarian tumor patient-derived samples, where a four-fold increase in autophagy-related markers between primary disease and dormant post-chemotherapy recurrence was detected [64]. Among the upstream signaling regulating autophagy, the inhibition of the PI3K/AKT/mTOR pathway and the activation of the energy sensor 5′adenosine monophosphate-activated protein kinase (AMPK) are generally observed in dormant cells [55,65]. In xenografted ovarian tumors, ARHI1 expression induces autophagy by inhibiting the PI3K/AKT/mTOR pathway and supporting nuclear localization of the transcription factor EB (TFEB) and Forkhead box O3 (FOXO3a), which in turn mediate the expression of crucial autophagy effectors [56]. mTOR-complex-1 (mTORC1) is the central intracellular hub for integrating autophagy-related signals. mTORC1 sustains cell growth and metabolic activity in the presence of nutrients (in particular amino acids, such as leucine, arginine, and glutamine) [66], growth factors, and high cellular energy levels, while inhibiting autophagy [67]. Contrarily, in energy deprivation conditions, low ATP/high AMP levels activate AMPK, an upstream, indirect mTORC1 inhibitor, thus promoting autophagy to enable stress adaptation and survival [68]. Interestingly, intracellular ATP levels are frequently decreased in dormant cells, driving AMPK activation [56,69]. A second mTOR complex, mTORC2, positively regulates mTORC1. Grow factors alone are sufficient to activate mTORC2, but its activity can also be modulated by several additional signals making this complex a highly versatile factor in sensing and transduction [70]. While the mTORC1 regulates autophagy directly, mTORC2 indirectly provides regulatory signals from insulin receptor phosphoinositide 3-kinase [71]. Notably, both the mTOR complexes play a central role in orchestrating metabolic reprogramming in cancer cells [67], and the alterations in mTOR-related signaling pathways probably dictate metabolic adaptations of dormant cells.

Coherently, dormancy correlates with the transcriptional up-regulation of driving autophagy-related genes, such as LC3, ATG4, ATG5, ATG7, and BECN1 [55].

Besides removing misfolded proteins, the regulation of mitochondrial functionality and the prevention of ROS accumulation are of extreme importance for the long-term survival of dormant cells. To this aim, dormancy is frequently associated with increased mitophagy as a major mechanism to eliminate damaged/dysfunctional mitochondria to manage oxidative stress balance [59,72]. Given the driving role of autophagy in supporting dormancy, targeting the autophagic activity with specific inhibitors, such as chloroquine and its derivatives, arises as a valid approach to impair dormant cell survival, reactivation, and metastatic capacity [55]. Pharmacologic or genetic inhibition of the autophagic process in dormant breast cancer cells induces the accumulation of dysfunctional mitochondria and oxidative stress, leading to cell death and, thus, limiting the dormant tumor cell population responsible for breast cancer recurrence [63].

The UPR is a multifaceted network of signal transduction pathways activated when the endoplasmic reticulum (ER) is functionally altered. UPR is triggered by the activation of three primary stress sensors: inositol-requiring protein 1 (IRE1), protein kinase RNA-like ER kinase (PERK), and activating transcription factor 6 (ATF6) [73]. These signal-transducing proteins detect the accumulation of unfolded proteins and activate adaptive processes through transcriptional and non-transcriptional responses, affecting every significant aspect of the ER-mediated secretory pathway [74]. Cancer cells are generally characterized by a high basal level of UPR activation, which provides survival advantages and supports oncogenic transformation by contributing to tumor growth, angiogenesis, and immune evasion [75]. During the acquisition of dormancy, UPR is frequently induced, as demonstrated by an overall increase in all the three major transducers of the UPR [54].

Moreover, the ERK/p38α balance plays a central role in modulating the dormant phenotype by inducing and integrating different signaling pathways in cancer cells [45]. In particular, p38 signaling protects dormant cancer cells from stress by inducing the UPR through the up-regulation of ER chaperone BiP and activation of PERK [46]. Low ERK1/2, in concomitance with high p38 activation, induces a G0-G1 cell cycle arrest through the up-regulation of different transcriptional factors, including p53, nuclear receptor subfamily 2 group F member 1 (NR2F1), and basic helix-loop-helix domain containing, class B, 3 (BHLHB3), and the down-regulation of FOXM1 and c-Jun, which promote G1 exit [47]. In turn, ERK1/2 is inhibited during dormancy induced by low expression of the urokinase-type plasminogen activator receptor (uPAR) [48,49]. Moreover, p38-dependent activation of ATF6α leads to the mTOR-mediated maintenance of basal survival in dormant cells. Under additional stresses, a low ERK/p38α ratio may also favor the expression of the chaperone BiP/Grp78, which inhibits Bax activation to avoid apoptosis and ensure adaptive survival to dormant cells [50]. Finally, in estrogen receptor-positive (ER+) breast cancer, the activation of the NF-KB pathway has been described to promote a dormant and metastatic phenotype [76].

#### 1.2.2. Extracellular Mechanisms

In addition to intrinsic pathways, cancer cells might enter the dormant state as a response to external signals from the tumor microenvironment (TME). The dynamic interaction between cancer cells and the TME plays a crucial role in modulating dormancy entry/escape [7]. This aspect is essential in the contest of metastatic dormancy, where the G0 cell cycle arrest is induced by the colonization of a new non-permissive microenvironment. Importantly, all the components of the TME play an active role in determining the success of DTCs in the metastatic site in a process that extends beyond the classical “seed and soil” hypothesis according to which seeds (cancer cells) only germinate in soils (tissues) passively providing the sources, nutrients, and environmental conditions they need [77]. Instead, cellular and structural components of the TME have an active role in determining the balance between dormancy and proliferation of colonizing cancer cells, finally determining their establishment [78].

Cancer-associated fibroblasts (CAFs) may influence cancer dormancy by producing and secreting a collection of different molecules, including TGFβ, interferons, interleukins, chemokines, and other cytokines [79]. In ER-breast cancer, CAF-secreted IFN-β drives a persistent activation of the IFN-β/IFNAR/IRF7 signaling axis in cancer cells, thereby favoring chemotherapy-induced dormancy [80]. In an ovarian cancer xenografts model, elevated IL-8, VEGF and IGF levels in the microenvironment are enough to rescue the ARHI-induced autophagy and drive toward a dormant phenotype [81]. In hormonal therapy (HT)-resistant breast cancer, in addition to the production of soluble factors, CAFs can also modulate cell dormancy through the transfer of extracellular vesicles containing their mitochondrial genome to cancer cells, thereby facilitating the exit from metabolic quiescence both in in vitro and in vivo models [82]. Interestingly, breast cancer cells located in the BM metastatic sites are able to prime mesenchymal stem cells to release exosomes containing specific miRNAs promoting dormancy and drug resistance in a subset of cancer cells [83].

Importantly, the composition of the immune population in the host tissue also plays a driving role in the modulation of the dormant phenotype in tumor cells. In a murine sarcoma model, an increased proportion of anti-tumor immune cells, especially natural killer cells (NKCs), compared to immunosuppressive cells, has been associated with the maintenance of tumor dormancy [84]. Indeed, perforin-mediated cytotoxicity, performed by NKCs, mediates dormancy by hampering malignant cells proliferation and forcing them to stay in a dormant state [85]. Moreover, effector T cells, mainly CD8+ and CD4+ T cells, contribute to the induction and maintenance of dormancy in different tumor models [86,87,88,89]. Noteworthy, the macrophages composition within BM may regulate the behavior of metastasizing breast cancer cells either by facilitating dormancy or reversing this state. In particular, the M2 macrophages form gap junctional intercellular communication with cancer cells, resulting in cycling quiescence. In contrast, M1 macrophages-released exosomes activate NFкB to reverse quiescent breast cancer cells to cycling metastatic cells [90]. Significantly, a sustained inflammation status may also influence dormancy in circulating tumor cells. For example, persistent lung inflammation is sufficient to awake disseminated dormant cancer cells into aggressively growing metastases [91].

Besides cellular components, the extracellular matrix (ECM) composition may also control the dormant status in cancer cells. Loss of integrin activation and fibronectin matrix re-organization are two driving processes that maintain dormancy by contributing to p38 activation and ERK inhibition [45,92]. Importantly, stromal cells also have a substantial ECM synthesis and remodeling capacity, further contributing to the ECM-mediated balance between dormancy and proliferation in cancer cells [93,94].

Among extracellular stimuli, hypoxia has been identified as a pivotal contributor to tumor dormancy [95]. Indeed, the hypoxia-mediated induction and stabilization of hypoxia-inducible factor 1α (HIF-1α) promote dormancy by inducing cell-cycle arrest and, simultaneously, cell survival by activating autophagy [96]. Hypoxic conditions in the primary tumor facilitate the establishment of dormant cellular subpopulations prone to metastasize. Primary lung cancer cells bearing activating epidermal growth factor receptor (EGFR) mutations enter a reversible dormant state under hypoxic conditions, resulting in resistance to EGFR tyrosine kinase inhibitor treatment and irradiation [97]. Furthermore, exposure to hypoxia increases the number of breast cancer stem cells (CSCs), and in agreement, inhibiting HIF-1α leads to tumor regression and delays tumor recurrence [98]. The hypoxic microenvironment in primary head and neck squamous cell carcinomas (HNSCC) and breast tumors induces the selection of a fraction of cells that concomitantly express hypoxia-related genes (GLUT1 and HIF-1α) and long-term dormancy genes, such as NR2F1, DEC2, and p27 [99]. Interestingly, hypoxic dormant cells up-regulate hypoxia markers in a reversible manner suggesting that the dormancy-like response is more long-lived than the hypoxic program. Indeed, post-hypoxic DTCs do not preserve high hypoxia gene expression while still maintaining the dormant phenotype, thus contributing to therapy resistance and metastatic dissemination [100]. Equally, defective angiogenesis, creating a low-oxygen milieu, has been correlated for a long time to a dormant phenotype in early primary tumors [101,102,103].

The accumulation of HIF-1α under normoxic conditions, a phenomenon known as “pseudohypoxia” [104], may also contribute to the switch between proliferation and dormancy. Indeed, several oncogenic mutations of growth factor receptors or downstream signaling molecules, such as EGFR, HER2, RAS, and BRAF, result in stabilization/activation of HIF-1α in different types of cancer [105]. Moreover, oncometabolites, such as 2-hydroxyglutarate, fumarate, and succinate, may accumulate due to mutations in isocitrate dehydrogenases, fumarate hydratase, and succinate dehydrogenase, respectively, thus inhibiting prolyl hydroxylase enzymes, which normally target HIF-1α for degradation, contributing to its stabilization [106]. Concordantly, elevated levels of succinate or fumarate stabilizing HIF-1α are able to reprogram breast cancer cells into a stem-like state [107]. However, the role of HIF-1α still remains controversial: while these findings demonstrate a key role for HIF-1α signaling in dormant cells, other evidence show that hypoxia can reactivate tumor growth, thus reverting cell dormancy at the metastatic site [108]. Future efforts will be necessary to clarify this opposite role of HIF-1α on regulating cell dormancy and re-awakening, with particular attention on the role exerted by the different microenvironmental composition between the primary tumor and the metastatic lesions.

Finally, the absence of proliferation-inducing stimuli may also lead to dormancy. In pancreatic cancer, the ablation of the oncogenic drivers mutant KRAS and c-MYC induces a dormant state, allowing the survival of a few residual cancer cells, which rely on autocrine IGF1/AKT as an adaptive mechanism [51]. Pancreatic cells surviving KRAS oncogene ablation show features of dormant cells, are responsible for tumor relapse, and mostly rely on oxidative phosphorylation (OXPHOS) for survival [52].

In conclusion, several factors may influence tumor cell entry in a dormant state, making this adaptive strategy particularly complex and multifaceted, but providing numerous possible strategies to prevent dormancy-related tumor relapse and metastatic disease.

### 1.3. Awaking Dormant Cancer Cells: A Leading Strategy for Tumor Success

One of the most intriguing aspects of dormant cells is their ability to interpret homoeostatic signals from the microenvironment. This feature allows cancer cells to evade immune surveillance and chemotherapy during the dormant phase and, on the other side, to reawaken in response to specific signals, re-acquiring proliferative and metastatic potential and finally re-establishing detectable tumor masses. However, despite the growing interest in these issues in recent years, stimuli inducing dormant cells awakening are still largely unknown, and further studies are needed to understand the molecular mechanism driving these phenomena.

Remodeling microenvironmental composition is one of the main drivers of cancer cell dormancy escape. Indeed, the transition from dormancy toward a proliferating state relies on the ability of tumor cells to adhere to ECM as demonstrated by the induction of dormancy escape following deposition of type I collagen at the metastatic location, promoting the reawakening of disseminated DCTs by inducing cytoskeletal reorganization and β1/β4-integrin signaling [109] or the formation of F-actin stress fibers dependent on integrin 1 phosphorylation of myosin light chain (MLC) [110]. Moreover, changes in the microenvironmental nutrient composition may alter the dormant/proliferation balance. For example, targeting pyruvate uptake in breast cancer-derived lung metastases impairs collagen hydroxylation and consequently tumor growth in the metastatic niche [111].

Another leading factor of dormancy’s escape is the presence of new vessels that may reactivate tumor growth in a process known as “angiogenic switch”. Indeed, it is well known that anti-angiogenic therapies may induce tumor dormancy, while angiogenic ones promote tumor rescue [112]. The pro-dormant effect of thrompospondin-1 is lost in sprouting neovasculature which has an opposite effect by releasing TGF-β1 and periostin as pro-proliferation factors favoring micrometastatic outgrowth during the awaking of quiescent cells [94]. Moreover, endothelial cells produce angiogenic factors, which elicit Notch3 activation in neighboring tumor cells and promote metastatic outgrowth [113].

Changes in the immune system composition, especially at the tumor immune microenvironment level, such as during ageing or following the treatment with chemotherapy and immunosuppressive drugs, may drive the escape from dormancy [114]. Moreover, inflammation and the wound healing process due to surgery may promote distant cancer outgrowth, while anti-inflammatory treatments reduce metastasis formation [115]. In a dormant-emergent metastatic breast cancer progression model, the quiescent cell population can be induced in a proliferative state following inflammatory stimuli, namely lipopolysaccharide plus EGF [116]. In mouse models, inflammation-derived proteases, neutrophil elastase, and matrix metalloproteinase-9 can induce proliferation in dormant cells by cleaving extracellular laminin and exposing a specific epitope that triggers an integrin-mediated signaling cascade in cancer cells, finally reactivating the proliferation of dormant cancer cells at sites of metastasis [91].

Understating signals and molecular mechanisms driving the escape of cancer cells from the dormant state is relevant for developing therapeutic approaches specifically targeting dormant cells, making them again sensitive to anti-proliferative drugs.

## 2. The Metabolic Landscape of Cancer Cell Dormancy

### 2.1. Metabolic Modulations Regulating the Shift between Proliferation and Quiescence

A strict connection exists between metabolism and cell cycle progression. Proliferating cells display higher biosynthetic requirements than non-proliferating cells to double in size and synthesize proteins, lipids, and DNA to generate a new cell. This makes cellular proliferation a metabolically demanding process, employing a large amount of biomass and energy. The commitment of a cell to proliferate is therefore strictly dependent on its metabolic status and nutrient availability [117]. Cell cycle progression is indeed interconnected through a reciprocal activation of metabolic enzymes and cell regulators [118], which cooperate with nutrient-sensors in modulating cell growth [119]. It follows that necessary metabolic adaptations characterize the shift between proliferation and quiescence.

Already from early studies, it emerged that cells display higher rates of glucose uptake and lactate production during the logarithmic growth compared to non-proliferating phases [120,121]. In accordance, quiescent cells frequently display an oxidative metabolism with reduced dependence on glycolysis and exogenous glucose consumption [52]. Before antigen stimulation, naïve T-cells are maintained in a quiescent state characterized by a G0 cell-cycle stasis, low transcriptional and translational activities, and a catabolic metabolism of glucose and amino acids. Quiescent naïve T-cells mainly produce ATP through OXPHOS [122], which is essential for preserving lysosome functionality and maintaining cellular homeostasis [123]. Utilizing an organotypic 3D tissue culture model in which non-transformed mammary epithelial cells are induced to a cell-cell and cell-matrix-mediated proliferative arrest, Coloff and collaborators found that specific alterations in glutamate metabolism characterize quiescent cells. Specifically, dormant cells display reduced glutamine consumption and transaminases level while exhibiting increased glutamate dehydrogenase expression, finally leading to decreased non-essential amino acid biosynthesis [124].

However, the metabolic slowdown is not a general hallmark of dormant cells. Primary human fibroblasts retain a high metabolic state when induced into quiescence, sustaining the breakdown and re-synthesis of lipids and proteins as a “self-integrity conservation” strategy and supporting ECM production. Under a dormant phenotype, these cells, thus, retain a high rate of glucose consumption, sustaining all the branches of central carbon metabolism activity, with a specific increase in the backwards fluxes from pentose phosphate pathway (PPP) to glycolysis and from α-ketoglutarate to citrate in the TCA cycle [125]. Similarly, adult stem cells, which are generally in a dormant-like status, are inclined to display metabolic profiles similar to those of proliferating cells, even though they are not dividing [126]. For example, hematopoietic stem cells, located in “stem cell niches” in the bone marrow, adapt to the low-oxygen regions they occupy by limiting the utilization of the electron transport chain (ETC) and primarily relying on glycolysis for ATP production [127,128]. This metabolic adaptation also provides an ROS-scavenging strategy in hematopoietic stem cells [129].

Therefore, dormancy is an adaptive strategy exploited by different organisms and cell types to survive harsh conditions or perform definite activities, each of which is characterized by specific metabolic requirements beyond the simple regulation of cellular proliferation and cell cycle progression. This implies that, although a G0 cell cycle arrest generally characterizes cell dormancy, this phenomenon is associated with different metabolic adaptations, depending on the cell type and on the purpose of the quiescence-entry strategy.

### 2.2. Metabolic Rewiring of Tumor Cell Dormancy: Critical Adaptations toward All the Steps of Cancer Progression

#### 2.2.1. Metabolic Adaptations Supporting Cellular Dormancy in Primary Tumors

For a long time, tumor growth was considered to be mainly sustained by the so-called “Warburg metabolism” (or “aerobic glycolysis”), characterized by an increase in glucose uptake and enhanced lactate production even in the presence of adequate oxygen concentration [130]. However, more recent studies underlined that cancer cells have heterogeneous and flexible metabolic preferences and that aerobic glycolysis is not a common feature across tumor types [131]. Such heterogeneity is influenced by both cancer cells intrinsic factors (such as genetic and mutational variability or differentiation status) or microenvironment-derived drivers (nutrient and oxygen availability and interactions with stromal cells and ECM) [132]. In this scenario, the importance of OXPHOS and other energy sources different from glucose in tumor growth and metastatic progression emerged [133,134,135,136,137,138,139]. Indeed, primary tumors contain proliferating, slow-cycling, dormant, and apoptotic cells and pre-metastatic tumor cells; each cell population is characterized by a specific metabolic profile supporting its distinctive status. In particular, tumor cells characterized by slow proliferation are, in most cases, dependent on mitochondrial respiration [140,141].

In accordance, increasing evidence supports the idea that subpopulations of quiescent cells within the tumor mass mostly rely on mitochondrial oxidative metabolism (Figure 1). Using an established lipophilic-dye (Vybrant^®^ DiD) retention model and performing whole-transcriptomic profiling (mRNA-Seq) of four different breast cancer subtypes, Quayle and collaborators identified a “quiescent breast cancer cell transcriptome” associated with late recurrence that emerged to be enriched with gene products involved in OXPHOS [142]. Similarly, Zhang and collaborators demonstrated that slow proliferative BRAFV600E melanoma cells, resistant to MAPK inhibitors, activate mitochondrial biogenesis and OXPHOS to meet their bioenergetics needs and survive [143]. Interestingly, La and collaborators showed that the proto-oncoprotein c-Myc, which usually drives malignant cell cycle progression, is expressed at relatively high levels in p27highKi67low quiescent melanoma cells, where it selectively transactivates OXPHOS-related genes, including subunits of isocitric dehydrogenase 3 (IDH3), while its binding to cell cycle progression gene promoters is reduced in quiescent cells [144].

Specific molecular pathways regulate the metabolic shift toward OXPHOS in dormant cells. AMPK is frequently up-regulated in quiescent cells, and it plays a crucial role in the shift toward oxidative metabolism through the expression/activation of PGC1α and the modulation of HIF activity in various tissues and cell types [65,145,146]. Interestingly, Hampsch and collaborators, by investigating an in vivo model of estrogen withdrawal-induced dormancy in ER+ breast cancer, demonstrated that high levels of AMPK promote mitochondrial respiration driven by FA oxidation. Actually, in in vivo settings, inhibiting AMPK or FA oxidation supports clearance of dormant residual disease [147]. AMPK is also a key player in regulating antioxidant defense during oxidative stress. Indeed, AMPK up-regulates several antioxidant genes [148], including the key regulator of the antioxidant response, the nuclear factor erythroid 2-related factor 2 (NRF2) [149], which has been recognized as a master regulator of cell dormancy in breast cancer [150]. Indeed, Fox and collaborators demonstrated that the small population of dormant breast tumor cells surviving Her2 down-regulation activates an NRF2 antioxidant transcriptional program in vivo. Interestingly, NRF2 activation also persists in recurrent tumors and in breast cancer patients with poor prognosis. Moreover, constitutive NRF2 activity also confers sensitivity to glutaminase inhibition, thus preventing the reactivation of dormant tumor cells in vitro [150]. Furthermore, despite the activation of glycolysis or OXPHOS, the AMPK stress response maintains a quiescent state by repressing cellular proliferation through mTOR inhibition [151].

Starting from this evidence, Zhang and collaborators proposed mitochondria as the “Achilles heel” of slowly proliferating tumor cells in nutritionally compromised microenvironments. In particular, they suggested targeting these organelles to eradicate non-proliferating tumor cells while saving non-proliferating cells in healthy tissues [152]. Promising results were obtained by using the mitochondrial OXPHOS inhibitor VLX600 to decrease the viability of quiescent cells in 3D colon cancer microtissues. The treatment with VLX600 forces quiescent cells to increase glycolysis to sustain their bioenergetic demand. However, this shift towards glycolysis is hampered by the low glucose content in the TME, resulting in a condition of limited plasticity and an inadequate ability to respond to decreased mitochondrial functionality, thus leading to bioenergetics catastrophe and tumor cell death [153]. Interestingly, a phase I study demonstrated that VLX600 is reasonably well tolerated in patients with refractory advanced solid tumors, paving the way for its possible future application as an anti-cancer agent [154].

Different nutrients can support mitochondrial metabolism besides glucose. Therefore, the metabolic dependency of dormant cells on mitochondrial metabolism could represent an additional adaptation under stressful conditions, as persisting cells can utilize different metabolic substrates to support their metabolism. In this line, Pascual and collaborators found that a subpopulation of slow-cycling CD44^bright^ cells in human oral carcinomas express high CD36 FA translocase levels and display overexpression of genes associated with lipid metabolism, ensuring them to rely on dietary lipids to promote metastasis. Indeed, CD36 translocase-expressing cells uptake FAs to fuel mitochondrial lipid β-oxidation [155]. Similarly, Nakayama and collaborators recently demonstrated that prostate dormant cancer cells show increased lipid metabolism, which induces protoporphyrin IX accumulation and, hence, sensitivity to photodynamic therapy (ALA-PDT). Treatment with triacsin C, a key inhibitor of lipid metabolism, reduces dormant cell cytotoxicity induced by ALA-PDT [156].

A different metabolic setting was described in dormant ovarian cancer cells overexpressing the autophagy inducer ARHI [56]. In these cells, ARHI expression promotes bioenergetics perturbation, oxidative stress, reduction of mitochondrial respiration, loss of mitochondrial mass, and decreased mitochondrial membrane potential, leading to an intense mitochondrial dysfunction. Moreover, an ARHI-mediated up-regulation of glucose and glutamine uptake was described. Indeed, glutaminolysis may serve as an adaptive response to oxidative stress by providing NADPH for ROS scavenging. Targeted inhibition of these metabolic pathways may represent a promising approach to overcoming ARHI-mediated tumor dormancy [59].

As described for proliferating tumor cells, the metabolic profile of dormant cells within the tumor mass is influenced by external stimuli derived from both cellular and microenvironmental compartments of the TME. PDAC cells exposed to hepatic stellate cells, simulating a physiological liver stroma, display an increase in the expression of succinate dehydrogenase subunit B (SDHB) and an elevated oxidative metabolism associated with the quiescent cell status; in contrast, exposure to hepatic myofibroblasts, which mimic liver inflammation, promotes reversal of this quiescent status, a proliferation boost, and a more glycolytic phenotype [157]. Moreover, environmental conditions, such as hypoxia, also drive metabolic adaptations in dormant tumor cells. Recently, Carcereri de Prati and collaborators demonstrated that hypoxia/HIFs-induced dormancy correlates with lower energy metabolism, characterized by reduced glucose consumption and lactate production, together with an increased autophagy program to promote cell survival [96]. Moreover, Sanchez Calle and collaborators proved that the expression of the long non-coding RNA NR2F1-AS1, whose over-activation induces the quiescence-like state in ER-positive breast cancer cells, leads to the enrichment of hypoxia and glycolysis pathways [158], in agreement with previous studies conducted in dormant hematopoietic stem cells [159].

Together, this evidence indicates that cells undergoing a dormant state in the primary tumor are generally characterized by an overall metabolic slowdown but display different adaptations to cope with their energetic and biosynthetic requirements according to tumor type and microenvironmental conditioning. Although this evidences may seem contradictory, they reflect a wide diversity in the metabolic program of dormant cells, which depends on nutrient availability, the tissue of origin, and the microenvironmental cues, leading to the impossibility of defining a unique metabolic setting associated with dormancy for different tumors. However, a common characteristic of tumor dormancy can be recognized in increased ROS withstanding, which can be achieved with different strategies depending on the tumor setting: through an efficient OXPHOS [52], diverting glycolytic intermediates towards glycerol metabolism and phospholipids [160], or activating autophagy and up-regulating glycolysis [59].

#### 2.2.2. Metabolic Adaptations Supporting Tumor Dormancy throughout the Metastatic Cascade

The ability to undergo a state of dormancy is essential during the metastatic cascade. Metastasis formation is a complex multistep process requiring different metabolic adaptations in cancer cells to survive the circulation, successfully colonize the metastatic niche, and finally establish a secondary tumor in distant organs [161].

Upon the invasion into the surrounding tissues, cancer cells undergo a state of dormancy associated with fundamental metabolic changes mainly aimed at supporting redox homeostasis and potentiate antioxidant defenses to survive the circulation (Figure 1). In melanoma, the oxidative stress limits tumorigenesis by DTCs, while the treatment with the antioxidant N-acetylcysteine (NAC) enriches the number of melanoma cells in the bloodstream and results in lymph node metastasis [162]. Indeed, anchorage-independent growth is accompanied by metabolic reprogramming, increasing mitochondrial NADPH levels to mitigate mitochondrial ROS. In low proliferating lung tumor spheroids, detachment from the ECM reduces glucose and glutamine oxidation and induces cytosolic isocitrate dehydrogenase-1-dependent reductive carboxylation of α-ketoglutarate to citrate, which then enters the mitochondria to produce NADPH for ROS mitigation [163]. Therefore, the *leitmotiv* of dormant DTCs is to reduce the detrimental oxidative stress, an adaptation that can be successfully achieved by slowing down the metabolic rate through an efficient OXPHOS. In keeping with this observation, an analysis of gene expression signatures in an orthotopic breast cancer model showed that DTCs increase mitochondrial respiration and PGC-1α expression to support mitochondrial biogenesis during the transit to target organs of metastasis. Accordingly, PGC-1α suppression significantly impairs mitochondrial biogenesis and OXPHOS, thus decreasing the number of DTCs and the frequency of metastasis [164]. By single-cell RNA sequencing, Dudgeon and collaborators recently defined a dormancy signature in a pancreatic cancer mouse model by analyzing genes and pathways differentially expressed in dormant DTCs, primary tumors, and reactivated clones. At the pathway level, the analysis identified as enriched in DTCs a cellular program of decreased proliferation, protein biosynthesis/expression, and cellular energy metabolic pathways, such as the TCA cycle, OXPHOS, and the PPP. Noteworthy, these results were further validated in human DTCs from patients undergoing surgery for localized pancreatic cancer [165]. Interestingly, the metastatic route may also influence the metabolic preferences of DTCs. According to the higher availability of oleic acid-containing vesicles in the lymph rather than in the blood, DTCs metastasizing through the lymphatic system display reduced ROS-induced ferroptosis thanks to an oleic acid-mediated decrease in cell membranes desaturation and a consequently reduced sensitivity to lipid peroxidation [166].

Once DTCs have reached a target tissue, they need to maintain the dormant status, allowing the acquisition of multiple adaptations to colonize the new environment. Importantly, the successful strategy strongly differs depending on the target tissues and the environmental characteristics (such as nutrients, ECM, and stromal cells composition) within different organs [167,168] (Figure 1). For example, pyruvate is enriched in the lung compared with the plasma [139]. This implies that breast cancer DTCs colonizing the lung adapt their metabolism to utilize pyruvate to establish in the new environment by increasing the production of α-ketoglutarate through mitochondrial alanine aminotransferase 2 (ALT2) conversion, supporting collagen deposition and remodeling [111] and potentiating mTORC1 signaling [169]. Differently, breast cancers colonizing the brain, due to the low availability of serine in the cerebral tissues, need to maintain high expression of the first enzyme of the serine biosynthetic pathway phosphoglycerate dehydrogenase (PHGDH) [170]. Moreover, acetate emerged as a leading energy source in brain metastases from a broad spectrum of origin tissues (such as breast, lung, kidney, and melanoma) in which primary tumors do not display a significant acetate uptake. This evidence suggests that the ability to oxidize acetate is a specific adaptation to the brain microenvironment [171]. A different strategy was observed in the peritoneum cavity, where omental adipocytes support energy metabolism of metastasizing ovarian cancer cells by providing FAs and by inducing the up-regulation of the membrane-associated FA binding protein CD36, facilitating FA uptake [172]. A crucial role of OXPHOS has been highlighted in breast micrometastases colonizing the lung. By using single-cell RNA sequencing and patient-derived-xenograft models of breast cancer, Davis and collaborators demonstrated that metastatic cells exhibit a distinct transcriptome program compared to primary tumor cells, characterized by OXPHOS as the top up-regulated pathway in the rare micrometastatic cells, while higher levels of genes associated with aerobic glycolysis are found in the primary tumor cells. Moreover, micrometastases show a distinct metabolic profile and higher levels of metabolites supporting OXPHOS, such as glutamine and glutamate. In keeping, lung dissemination is strongly impaired by pharmacological inhibition of OXPHOS, suggesting this metabolic pathway is a potential therapeutic target to halt the metastatic process [173]. Accordingly, a new Flura-seq technique identified a strong up-regulation of mitochondrial Complex I-encoding genes in lung micrometastases derived from breast cancer. This adaptation is correlated with increased oxidative stress and the activation of counteracting antioxidant programs, including the up-regulation of NRF2-driven genes to detoxify ROS. Interestingly, this gene signature is specific for the metastatic location, suggesting that it is a distinctive adaptive response to oxidative stress in lung micrometastases. Indeed, this transcriptome is lost upon removing cancer cells from the tissue microenvironment and placing them in culture, indicating a crucial role of the local *milieu* in promoting metabolic adaptation of the metastatic cell [174]. Coherently, breast cancer cells colonizing the lung increase PGC-1α activity to support metabolic flexibility and ATP production through OXPHOS [175]. Similar metabolic requirements were described in liver micrometastases of tumor-bearing KPC mice, a model of pancreatic ductal adenocarcinoma, which show a more robust SDHB expression than macrometastases, indicating an increase in mitochondrial metabolism [157]. Similarly, the small subpopulation of oral squamous cell carcinoma cells metastasizing the lymph nodes are characterized by a slow-cycling phenotype and increased FAs uptake to sustain oxidative metabolism in the mitochondria [155].

Moreover, RNA seq analysis, direct metabolite profiling, and in vivo [U-^13^C]-glucose tracing experiments demonstrated an increased OXPHOS in melanoma brain metastases compared to primary melanomas and extra-cranial metastases. Treatment with the OXPHOS inhibitor IACS-010759 reduces the risk of brain metastasis development in a mouse model [176].

Growing evidence describes the metabolic flexibility of cancer cells that undergo metastatic travel depending on the type, location, and stage of the primary tumor and the nutritional composition of the respective sites of metastasis. This great heterogeneity in metabolic adaptations makes quiescent metastatic cells particularly difficult to target and eradicate. A deeper understanding of the complexity of these processes is needed to enable a practical approach against dormant metastatic cells.

#### 2.2.3. Metabolic Adaptations of Dormant CSCs

The term CSCs defines the cellular subpopulations within the tumor mass possessing unique stem-like features, such as self-renewal capability, differentiation into multiple cell types, and high resistance to stress and apoptosis [177]. CSCs play a key role during tumor initiation, progression, and metastatic relapse. Moreover, they are mainly responsible for therapy failure and poor clinical outcomes. Traditionally, CSCs have been considered a very small subpopulation of quiescent cells; however, recent evidence demonstrated that CSCs can be relatively abundant and switch between dormant and proliferating states [177]. CSCs share some features and signaling pathways with dormant cancer cells, including therapy resistance and the ability to metastasize and evade the immune system, suggesting that CSCs and dormant cells are two sides of the same coin [15]. Nonetheless, not all CSCs are dormant, even though stem cells display more dormant properties than non-stem cell counterparts [177]. Equally, not all dormant cells can be classified as CSCs [178]. Indeed, dormant cancer cells likely include both CSC and non-CSC subpopulations [13], and CSCs do not necessarily show dormant-like features due to their ability to exchange between dormancy and proliferation. Based on this property, CSC can be classified into: (i) dormancy-competent CSCs (DCCs), (ii) dormancy incompetent CSCs, (iii) cancer repopulating cells (CRCs), and (iv) DTCs [14]. DCCs supply the tumor subpopulation of CSCs able to switch between dormancy and rapid growth. Conversely, dormancy-incompetent CSCs, usually enriched in advanced diseases, are characterized by the accumulation of somatic mutations and have, therefore, lost their ability to enter dormancy. CRCs are responsible for self-renew post-therapy and foster the metastatic onset. Finally, DTCs are CSCs, either displaying stem-like or differentiated features, located in distant secondary organs or the bloodstream. DTCs are usually dormant but preserve the ability to reawaken and fuel metastatic outgrowth [14,179]. Overall, dormant CSCs are a relevant source for disease recurrence. Characterizing CSCs markers and metabolic adaptations will enable the isolation of dormant tumor cell populations as therapeutic solutions for unmet medical needs. Interestingly, specific markers/factors characterize CSCs and dormant cells, providing useful diagnostic targets to direct and focus therapeutic strategies in the future (Table 1).

The metabolic profile of dormant cells reflects the function they perform during tumorigenesis. In this line, data are still controversial about the differences between the metabolic states of quiescent CSCs and their differentiated progeny [180]. Indeed, although some studies report, as mentioned above, the dependence of CSCs quiescence on OXPHOS [181,182,183], some others describe a greater dependence on anaerobic glucose metabolism [184,185,186]. In dormant brain tumor stem cells (BTSCs), the activation of a metabolic process related to lipid alteration emerged as a specific signature for dormancy. Indeed, these cells express glycerol-3-phosphate dehydrogenase 1 (GPD1), which generates the glycerol-3-phosphate, a precursor for the glycerophospholipid metabolism. In BTSCs, GPD1 expression activates cellular stress response-related pathways, leading to a decrease of intracellular ROS similar to that observed in cells exhibiting efficient mitochondrial OXPHOS [160]. Leukemia stem cells (LSCs) overexpressing BCL2 are metabolically dormant and characterized by relatively low levels of ROS. Therefore, these cells depend more on mitochondrial respiration than glycolysis to meet their energy demands and survive. In keeping, BCL-2 inhibitors ABT-737 and ABT-263 hinder oxidative respiration and promote the selective eradication of ROS-low LSCs [182]. Similarly, radioresistant glioma stem cells depend more on OXPHOS than differentiated glioma cells [181], and the quiescent subpopulation of stem cell-enriched chronic myeloid leukemia cells possess increased mitochondrial oxidative functions compared to normal hematopoietic stem cells [66].

Differently, in colorectal cancer models, CD133+ CSCs exhibit increased expression of the glycolytic and TCA cycle pathways components, together with a drastic increase in nucleoside mono- and diphosphate levels. Moreover, colorectal CSCs also display a strong intracellular accumulation of free amino acids, suggesting a drastic inhibition of protein synthesis [187]. In triple-negative breast cancer cells, the treatment with histone deacetylase (HDAC) inhibitors leads to a reprogramming of differentiated cells into stem-like cells. This treatment promotes enhanced PPP for the production of cellular NADPH and intracellular ROS detoxification [188].

Besides relying on either glycolysis or OXPHOS, emerging evidence proves that glutamine is a crucial energy source for CSCs [189,190]. In non-small-cell lung carcinoma and pancreatic cancer cells, decreasing glutamine availability lowers intracellular levels of reduced GSH, with a consequent ROS accumulation, ultimately resulting in a decreased proportion of intratumoral CSCs in vivo [189]. Coherently, inhibition of glutaminase 1, the enzyme that catalyzes glutamate formation from glutamine, suppresses the expression of the stemness markers in hepatocellular carcinoma and primary and metastatic head and neck cancer tissues [190,191].

Recent insights into the hallmarks of CSCs suggest that merely targeting stemness features is not sufficient to eliminate the CSCs population and prevent tumor relapse. However, targeting the metabolic adaptations supporting the quiescent status of CSCs could be a more powerful strategy than inhibiting the CSCs stemness directly to overcome drug resistance and tumor relapse.

#### 2.2.4. Metabolic Adaptations of Therapy-Induced Dormant Cells

Cancer therapy often determines the emergence of drug-resistant clones, eventually leading to therapy failure. While some tumors are intrinsically insensitive to anti-cancer therapies due to pre-existing resistance factors (intrinsic resistance), others become resistant during drug treatment (acquired resistance) [24]. Therapy-induced resistant cells are generally characterized by transient slow proliferative/quiescent status, even maintaining their viability. This dormant condition is strictly dependent on the presence of the cytotoxic stimuli and may be reverted following drug withdrawal. Moreover, the dose and duration of treatment determine whether tumor cells become dormant or senescent or trigger pro-apoptotic pathways [217]. Noteworthy, drug-resistant cells are not only present within the primary tumor but are also in distant organs as DTCs [218]. Thus, identifying metabolic liabilities of therapy-induced dormant cells is of primary importance for preventing primary tumor relapse after therapy and metastatic disease.

Some studies showed that chemotherapy-resistant cells mostly rely on OXPHOS in keeping with the selection of a quiescent subpopulation (Figure 1). For example, treatment of melanoma cells with different chemotherapeutics promotes the enrichment of slow-cycling long-term tumor-maintaining cells expressing the H3K4-demethylase JARID1B/KDM5B/PLU-1. These cells are characterized by an increase in OXPHOS and mitochondrial metabolism. In agreement, the inhibition of mitochondrial respiration sensitizes melanoma cells to therapy, independently from their genotype [22]. Moreover, in the gastrointestinal stromal tumor (GIST), Imatinib treatment promotes tumor cell quiescence and shifts the metabolism of these cells from glycolysis to OXPHOS. Accordingly, Imatinib treatment impairs the recruitment of glucose transporters to the cell surface, resulting in a dramatic decrease in glucose uptake [219]. Conversely, Sansone and collaborators demonstrated that neoadjuvant HT in metastatic breast cancer favors the enrichment of dormant CD133^high^/ER^low^ therapy-resistant cells characterized by defective mitochondrial oxidative metabolism and autocrine IL-6 production. Interestingly, chronic treatment of these resistant cells raises the levels of paracrine IL-6 expression and fosters the exit from quiescence via an IL6/Stat3/Notch3-mediated induction of mitochondrial activity. IL6/Stat3 blockade by Tocilizumab restores HT sensitivity in resistant cells and suppresses the generation of CD133^high^ cells, thus preventing the self-renewal of resistant CD133^high^ CSCs [220]. The same authors demonstrated that the shift toward OXPHOS responsible for dormancy escape and tumor re-growth is mediated by a horizontal transfer of mitochondrial DNA from CAF to HT-resistant cells via extracellular vesicles both in vitro and in vivo [82].

Despite these signs of progress, up to now, very little is known about the metabolic alterations underpinning the acquisition of a drug resistant-dormant phenotype. The difficulties in defining an overall therapy-induced dormancy metabolic profile are due to the heterogeneity of the metabolic patterns achieved by dormant cells to overcome the specific mechanism of action of cytotoxic agents they are exposed to.

### 2.3. Different Metabolic Strategies Exploiting Tumor Escape from Dormancy

Alongside the aforementioned plethora of intracellular and extracellular stimuli governing cancer cell quiescence, the activation of specific metabolic programs may favor the escape from a dormant phenotype in tumor cells. Interestingly, La Belle Flynn and collaborators determined a key role for the glycolytic enzyme 6-phosphofructo-2-kinase/fructose-2,6-biphosphatase 3 (Pfkfb3) in driving the balance between dormancy and awake in breast cancer cells. Indeed, dormant cells display a Pfkfb3^low^Autophagy^high^ phenotype. In contrast, a transition to a Pfkfb3^high^Autophagy^low^ phenotype is observed during metastatic relapse, suggesting that Pfkfb3, and hence, the glycolytic metabolism, play a critical role in promoting the escape of cancer cells from dormancy and reactivating proliferative programs [62]. Otherwise, Shinde and collaborators demonstrated that the growth of breast cancer cells disseminated to the lung relies on the activity of pyruvate carboxylase, the enzyme that converts pyruvate into oxaloacetate to sustain gluconeogenesis and to replenish the TCA cycle. Interestingly, this requirement is specific to the pulmonary district, while it is not necessary for the extra-pulmonary tumor initiation, reflecting the relevance of the specific metastatic microenvironment composition in defining metabolic adaptations [221]. Other evidence points to the relevance of lipid metabolism, particularly lipid β-oxidation, during metastatic re-growth [155]. In a breast cancer organoid culture model, cells that survive tumor regression and sustain tumor relapse exhibit altered lipid metabolism, elevated ROS levels, and increased oxidative DNA damage driving the residual cells toward tumor recurrence. Indeed, both NAC-dependent ROS scavenging and inhibition of hormone-driven proliferation of the mammary epithelium lead to the attenuation of tumor outgrowth [222].

In conclusion, despite the increasing knowledge accumulated in recent decades about the metabolic pathways sustaining cancer progression, so far, very little is known about the metabolic adaptations supporting dormant cells awake. Data describing the metabolic profile of awaked tumor-initiating cells are still limited, likely due to the lack of adequate experimental models to study this phenomenon. Exploring the metabolism of these cells will be helpful to underscore pathways supporting tumor re-growth, paving the way for the development of targeted therapies for the clinical benefit.

## 3. Novel Approaches to Eradicate Dormant Cells

Strategies to eradicate tumor cells rely on the ability of anti-cancer therapies to affect cell proliferation. Otherwise, dormant quiescent cells may be targeted through three different approaches: (i) maintaining cell dormancy to keep cells inactive and not proliferating; (ii) “awakening” or inducing the proliferation of dormant slow-cycling resistant cells, thus allowing the use of anti-proliferative drugs; (iii) eradicating dormant cells while they are still quiescent (Figure 2).

The first strategy is based on dormancy maintenance to elude tumor re-growth. This approach is mainly centered on activating the p38 MAPK pathway and inhibiting the ERK1/2 pathway to guarantee the preservation of anti-proliferative features by cancer cells [223]. This methodology could be promising but challenging due to the need for persistent treatments; therefore, it is associated with resistance phenomena. While the combined inhibition of ERK and Src kinase signaling prevents the metastatic outgrowth of dormant breast cancer cells by reducing the number of viable dormant cells, the solely pharmacological inhibition of SFK signaling or Src knockdown results in the inhibition of the dormant-to-proliferative switch [224]. Similarly, the CDK inhibitor Palbociclib [225] or the NR2F1 induction [41] promote the maintenance of a dormant state in cancer cells. Moreover, epigenetic alterations may retain the quiescence state in cancer cells. In hematological and epithelial tumor cells, low-dose of Azacytidine, which sustains changes in gene expression, may be used for reprogramming tumor cells into a non-proliferative state [226]. Furthermore, in uveal melanoma, HDAC inhibitors may be used to reprogram cells into a dormancy-like phenotype [227].

Differently, the awakening of dormant quiescent cells is expected to sensitize the newly cycling cells to anti-proliferative therapy. However, this approach is still controversial, since the reactivation of cell proliferation could worsen the tumor aggressiveness. For instance, in an HNSCC model, a MAPK p38α/βT-dependent dormancy phenotype is induced by TGF-β stimulation. Systemic inhibition of TGF-β-RI or p38α/β activities awakens dormant cells and increases the metastatic burden in the liver, spleen, and bone marrow [200]. However, depending on the cell context, different stimuli can induce cell cycle re-entry for the subsequent elimination of cancer cells with conventional chemotherapy. In acute myeloid leukemia (AML) mouse models, quiescent chemotherapy-resistant AML stem cells can be induced to enter the cell cycle by treatment with granulocyte colony-stimulating factor (G-CSF), which then increases sensitivity for the chemotherapeutic Cytarabine (AraC) [228]. Interestingly, in AML, an in vivo triple combination therapy strategy based on the G-CSF, checkpoint kinase 1 inhibitor (CHK1i), and AraC administration, enhances drug cytotoxicity compared to the treatment with AraC+CHK1i alone. Indeed, cells forced to enter the cell cycle by G-CSF stimulation become sensitive to the AraC+CHK1i treatment [229]. Similarly, neutralizing osteopontin, which allows dormant cells to re-start proliferating, synergizes with AraC to eradicate tumor cells [230]. Interfering with components of the ECM may be helpful to increase sensitivity toward chemotherapy further. In this context, the inhibition of the tyrosine kinase AXL, which usually maintains dormancy of myeloma cells by contact-dependent interactions with osteoblastic metastatic niches, induces the awakening of tumor cells [231]. Interestingly, combined approaches could also be valuable to overcome therapy-induced dormancy. For example, Imatinib induces GIST cell quiescence in vitro through the APC(CDH1)-SKP2-p27Kip1 signaling axis. The escape of GIST cells from quiescence through the inhibition of this pathway promotes proliferation and enhanced Imatinib-induced apoptosis [232].

A third, different approach could be eradicating cancer cells while still in the dormant state by exploiting their quiescence status. To this end, several strategies can be pursued based on the specific biology of dormant cells. Bcl-2/Bcl-XL inhibitors, such as ABT-737, effectively eliminate quiescent/slow-proliferating cells expressing high levels of anti-apoptotic proteins [233]. As previously mentioned, autophagy activation is a critical mechanism for DTCs maintenance. In this context, inhibition of autophagy could be a worthy approach to decreasing dormant cells’ survival and, hence, impairing their re-growth. In agreement, inhibition of autophagy-related 7 (ATG7) promotes dormant breast cancer cell death and reduces the total lung metastatic burden [63]. Moreover, the interaction between cancer cells and the microenvironment may be a further target for eradicating dormant cells. Antibodies against VEGF, IL-8, and IGF-R1 eliminate dormant ovarian cancer xenografts by enhancing cell death [81]. Moreover, epigenetic approaches may be exploited to remove dormant cells according to their peculiar epigenetic profile. Inhibition of epigenetic enzymes has proven successful results. Vinogradova and collaborators demonstrated that treating cancer cells with a specific inhibitor of the histone demethylase KDM5A results in the ablation of a subpopulation of founder cancer cells in multiple tumor cell line models [234]. In this view, several clinical trials aimed at reducing the reservoir of dormant DTCs and hence decreasing cancer relapse are under investigation in breast and prostate cancer patients (NCT00248703, NCT03032406, NCT01779050, NCT03572387).

The metabolic heterogeneity of dormant cells makes specific metabolic approaches feasible to eradicate quiescent tumor cells. In addition to the aforementioned VLX600, different metabolic drugs, already tested or approved as anticancer treatments, could represent additional therapeutic options. In particular, the frequently observed OXPHOS-dependent dormant cells could be targeted with mitochondrial respiratory chain inhibitors such as Metformin or Phenformin in various tumor types. In contrast, an approach based on glycolytic inhibitors for the dormant cells which rely on the Warburg metabolism could be pursued. In addition, other metabolic pathways, such as those involving glutamine, lipids, and autophagy, are potentially targetable in dormant cancer cells, mainly based on the recent success of the use of glutaminase and lipid transporter inhibitors or autophagy blockers [235].

In this perspective, preclinical studies show the efficacy of targeting metabolic pathways to fight dormant cancer cells through the three mechanisms mentioned above (Table 2). For example, glutaminase inhibition blocks the reactivation of dormant tumor cells and prevents recurrent tumor cell growth [150]. In different tumor models, inhibiting FA oxidation [147] or OXPHOS [152,173,176] slow-cycling cancer cells are eradicated, thus preventing tumor relapse. In this light, a very recent paper by Ting La and collaborators demonstrates that quiescent melanoma cells rely on c-myc-driven OXPHOS for survival. Treatment with the OXPHOS inhibitor IACS-010759 reduces cell viability in quiescent cells, whereas it does not significantly affect the viability of cycling cells; in agreement, silencing c-myc or the c-myc transactivated subunit of IDH3 preferentially eradicate quiescent cells, recapitulating the effect of the treatment with OXPHOS inhibitors [144]. Moreover, Havas and collaborators showed that inhibiting FA metabolism and the consequent ROS detoxification attenuates the somatic mutation-driven tumor recurrence of breast cancer [222]. Studies from Ornelas and collaborators suggest a different metabolic strategy to eradicate autophagic dormant ovarian cancer cells by targeting the up-regulation of glycolysis and glutaminolysis [59]. In support of the value of therapeutic approaches targeting metabolism, the inhibition of the critical metabolic enzyme GPD1 has been shown to prolong the survival of mice affected by glioblastoma, compromising the persistence of the dormant BTSC cells [160]. As previously mentioned, the small compound VLX600, interfering with mitochondrial respiration, effectively reduces the survival of dormant colon cancer cells. This molecule, to our knowledge, is the first drug targeting metabolism to enter phase I in patients with refractory advanced solid tumors [153,154].

The knowledge of the metabolic features governing cancer cell dormancy entry/escape may be significant in developing effective targeted therapies. Even though, currently, it is premature to hypothesize therapeutic treatments based exclusively on a metabolic approach, the use of metabolic drugs could be of great benefit if used in combination with traditional therapies.

## 4. Conclusions

Cancer dormancy represents an adaptive strategy that accompanies and sustains all the steps of tumor progression, allowing the survival of cancer cells in the hostile conditions they need to deal. The metabolic adaptations achieved by dormant tumor cells depends on the characteristics of the tissue of origin, environmental nutrient availability, and stromal influence, but share the common feature of slow-proliferating cells. In this context, the metabolic changes in dormant cells are often directed towards OXPHOS and ROS scavenging to ensure a sufficient energy supply under a low proliferation rate. The addiction of dormant tumor cells to specific metabolic pathways represents a helpful druggable weakness for therapeutic purposes. Future investigations may confirm that altered energy metabolism represents an attractive vulnerability to hinder different steps of tumor progression, including the maintenance of cancer dormancy or tumor relapse.

## Figures and Tables

**Figure 1 cancers-14-00547-f001:**
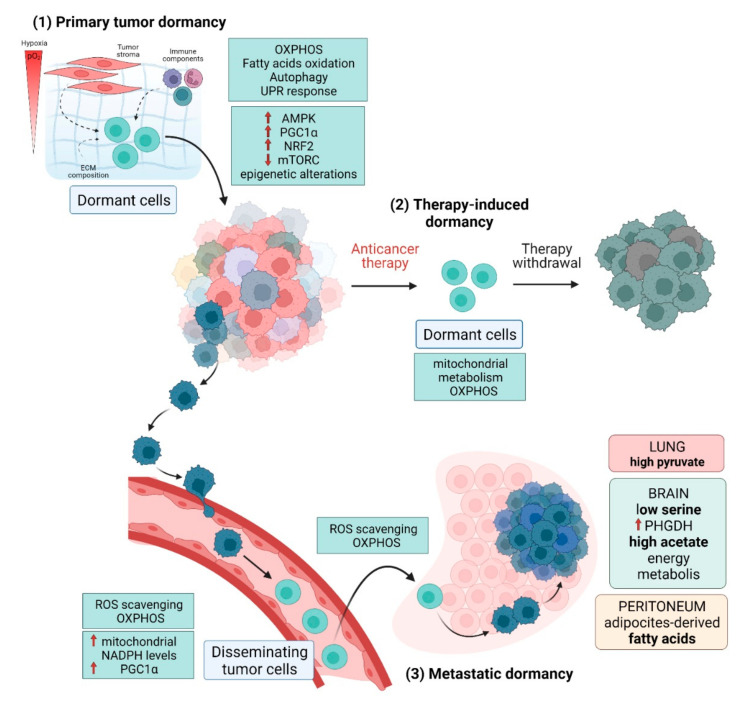
Cancer dormancy accompanies different steps of cancer progression. (1) Dormancy may occur during the early phase of malignant transformation by the action of acellular factors, such as ECM stiffness, oxygen availability, nutrient and growth factor composition, and cellular components, such as stromal, immune, and inflammatory cells. Quiescent cancer cells mainly depend on OXPHOS and lipid oxidation for their survival. (2) Chemotherapy treatment and ionizing radiation often fail to completely eradicate tumor mass, giving rise to a subpopulation of dormant cancer cells that survive anti-cancer therapies and are able to re-grow, thus causing tumor relapse even after a long time. (3) Dormancy may occur during the dissemination of tumor cells endowed with increased antioxidant capacity and mitochondrial metabolism. The metabolic adaptations supporting tumor re-growth mostly depend on the different nutrient availability in the metastatic location. Red arrows in figure show the activation/expression levels of different molecules involved in dormancy regulation.

**Figure 2 cancers-14-00547-f002:**
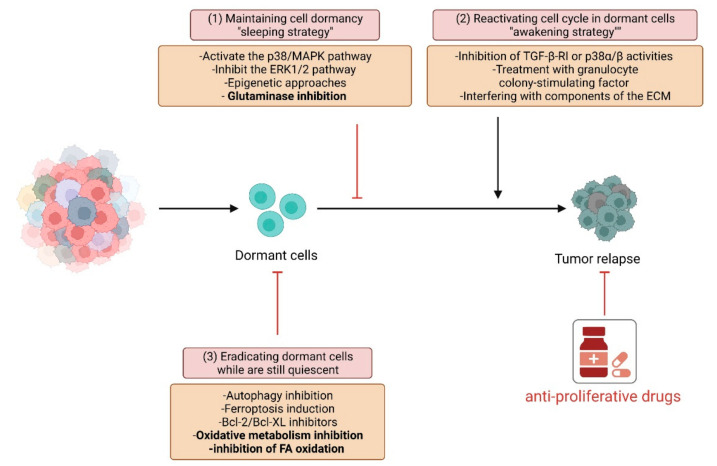
Strategies to eradicate cancer cell dormancy. Different approaches are proposed to target dormant cancer cells: (1) maintaining cell quiescence to avoid tumor re-growth promoting pro-dormancy pathways and inhibiting proliferative ones; (2) awaking dormant cells to sensitize them to anti-proliferative therapies—this strategy may be pursued by stimulating cells with different cell cycle re-entry inducers and targeting re-cycling cells with conventional anti-tumor drugs; (3) eliminating dormant cells while they are still quiescent targeting autophagy and/or OXPHOS metabolism.

**Table 1 cancers-14-00547-t001:** Molecular cues sustaining cell dormancy and specific markers of CSCs in different tumor types.

Tumor Type	Metastatic Site	Dormancy Factor	Mechanism	Ref	CSC Markers	Ref
Breast	Lung	Fbxw7	Increased levels	[34]	CD44^+^CD24^−^/^low^ ALDH^+^	[192,193]
	Lung	ATG7	Increased levels	[63]		
	Lung	CXCR4	Decreased levels	[194]		
	Lung	OXPHOS	Activation	[142,164,173]		
		ARHI	Increased levels	[58]		
		IFN-β/IFNAR/IRF7	Activation	[80]		
		NRF2	Activation	[150]		
	Bone marrow	LIFR	Increased levels	[108]		
	Bone	MSK1	Increased levels	[195]		
	Multiple sites	IKKβ	Activation	[76]		
	Intraperitoneal	KiSS1	Increased levels	[35]		
	Lung	NR2F1/DEC2/p27	Increased levels	[100]		
		AMPK/OXPHOS	Activation	[147]		
HNSCC	Lymph nodes	PRRX1	Increased levels	[196]	CD44^+^ ALDH^+^ BMI-1	[197,198,199]
	Bone marrow	NR2F1/NANOG	Increased levels	[41,99]		
	Bone marrow	TGFβ2	Increased levels	[200]		
Melanoma	Lung	KiSS1	Increased levels	[37]	ABCB5^+^ CD20^+^ CD271^+^	[201,202,203]
		OXPHOS	Activation	[143,144]		
Ovarian		KiSS1	Increased levels	[36]	CD44^+^, CD117^+^ CD133^+^ SP	[204,205,206]
	Intraperitoneal sites	MKK4	Increased levels	[207]		
		ARHI	Increased levels	[56,57]		
Prostate		Lipid metabolism	Activation	[156]	Sca1^+^, CD133^+^ CD44^+^ A_2_β_1_ ^hi^	[208,209]
	Bone marrow	TBK1	Increased levels	[72]		
	Bone	BMP-7	Increased levels	[210]		
	Bone	Wnt5a	Increased levels	[211]		
	Liver, Lymph node, Bone	GAS6/AXL	Increased levels	[212]		
	Bone	GDF10/TGFβ2/TGF-βRIII	Increased levels	[213]		
	Bone marrow	NR2F1/NANOG	Increased levels	[41]		
Pancreas	Liver and Lung	KRAS/C-Myc, IGF1/AKT	Activation	[51]	CD44^+^, CD24^+,^ ESA^+^, CD133^+^	[214,215,216]
		OXPHOS	Activation	[157,165]		

**Table 2 cancers-14-00547-t002:** Drugs targeting metabolic adaptations of dormant tumor cells.

Drug	Mechanism of Action	Cancer Model	Ref
VLX600	Iron chelator designed to interfere with intracellular iron metabolism, leading to mitochondrial OXPHOS inhibition	Colon cancer	[153]
VLX600	Iron chelator designed to interfere with intracellular iron metabolism, leading to mitochondrial OXPHOS inhibition	Phase I study on patients diagnosed with advanced solid tumors	[154]
Triacsin C	Inhibitor of lipid metabolism	Prostate cancer	[156]
CB-839 and BPTES	Glutaminase inhibitors. Inhibit the conversion of glutamine to glutamate, thereby limiting glutamate available for anaplerosis	Breast cancer	[150]
Dorsomorphin	AMPK inhibitor	ER^+^ Breast cancer	[147]
Ranolazine	FA oxidation inhibitor	ER^+^ Breast cancer	[147]
Etomoxir	FA oxidation inhibitor	ER^+^ Breast cancer	[147]
Perhexiline	FA oxidation inhibitor	ER^+^ Breast cancer	[147]
Oligomycin	ETC complex V inhibitor, OXPHOS inhibitor	Breast cancer metastasizing the lung	[173]
IACS-010759	ECT complex I inhibitor, OXPHOS inhibitor	Melanoma brain metastases	[176]
IACS-010759	ECT complex I inhibitor, OXPHOS inhibitor	Acute Myeloid Leukemia	NCT02882321
IACS-010759	ECT complex I inhibitor, OXPHOS inhibitor	Lymphoma	NCT03291938

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
