# Peer review of "Metabolic Features of Tumor Dormancy: Possible Therapeutic Strategies"

_cancers, 2022, doi:10.3390/cancers14030547_

Round 1
Reviewer 1 Report
This manuscript is a very informative review on metabolic features of tumour dormancy and their potential role in therapeutic failures.
It summarizes the potential mechanisms driving tumour cell dormancy and the awaking. Additionally, the authors follow the environmental and metabolic adaptation during different stages of tumour development concentrating on dormancy and site/therapy specific alterations.
The authors have compiled and reviewed numerous reports, and articles. Two figures were added which help to follow the subjects. However, there are some interesting points which would be important to mention in their manuscript. It would be very important to clarify the differences between dormant-quiescent and stem cell properties and it would be very nice if the authors summarized the possible dormancy markers of the main tumour types (breast, lung, colon cc. etc.) in one figure or table. Because there are many contradictions and overlapping marker sets in different studies which could contribute to the different interpretations and described metabolic adaptation mechanisms in cancer tissues.
I only have a couple of minor points to raise
I suggest clearing or mentioning some points
- the differences between cancer stem cells vs. dormant cells
- Is there any difference between quiescent vs. dormant cells?
- Which markers could help to distinguish cancer stem cells vs. dormant cells/therapy induced dormant cells in tumour tissues/tissues, in metastatic niches?
- the importance of metabolic plasticity in tumor ecosystem in correlation with dormant survival (I suggest highlighting the role of different mTOR complexes in metabolic adaptation a bit more – regulating autophagy and cellular survival…)
- oncogenic alteration induced pseudohypoxia was not mentioned in the manuscript
Some additional questions
- The authors discussed that hypoxia can induce dormancy (page 5 line 228-229). How could you explain the OXPHOS metabolic shift in dormant cells and that dormant cells mostly rely on OXPHOS for survival (page 6 line 258-259)?
- Which known drugs can be tested to target metabolic alterations in dormant tumor cells, as an additional option?
Finally, it is a great work that is very well structured and excellently written. I recommend to ask for minor revision prior final acceptance.
Author Response
Reviewer 1
We thank the reviewer 1 for his/her comments that allow us to ameliorate the quality of the manuscript.
- The reviewer suggested clarifying the differences between cancer stem cells and dormant cells. We agree with the reviewer that the differences and the shared features between CSCs and dormant cells are of primary importance in defining these cellular subpopulations. Therefore, we included a description of the specific characteristics of CSC and their classification based on their dormant state at the beginning of the paragraph regarding the metabolic adaptation of dormant CSC (lanes 795-827 of the “track-changes” file). Moreover, we included a table (Table 1) summarizing the major markers/factors characterizing CSCs and dormant cells.
- The reviewer asked to clarify the difference between quiescent and dormant cells. The terms dormancy and quiescence are often used interchangeably, even if some authors have classified dormancy as having a more profound, persistent arrested state than quiescence. Therefore, we now included this clarification following reviewer requests at lanes 58-69 of the “track-changes” file.
- The reviewer suggested to point out which markers could help to distinguish cancer stem cells from dormant cells/therapy induced dormant cells in tumour tissues/tissues and in metastatic niches Considering the importance of this aspect, we now included a table summarizing the main markers/factors characterizing CSCs and dormant cells (Table 1), and we introduced this issue at lanes 803-827 of the “track-changes” file.
- The reviewer suggested highlighting a bit more the role of different mTOR complexes in metabolic adaptation regulating autophagy and cellular survival. We agree with the referee that, although it plays a central role in modulating autophagy, we did not underline the importance of the mTOR pathway in the previous version of the manuscript. Therefore, we now included an insight about mTORC1 and mTORC2 underlying their importance in modulating the autophagic response and cancer metabolism (lanes 231-248 of the “track-changes” file).
- The reviewer noticed that oncogenic alteration induced pseudohypoxia was not mentioned in the manuscript. We agree with the reviewer that this is an important aspect to be mentioned over the text. Therefore, we now included a brief description of the oncogenic alterations and oncometabolites driving pseudohypoxia in cancer cells and their possible implication in the induction of a dormant stem-like state (lines 395-406 of the “track-changes” file).
Additional questions:
- The reviewer brought the attention to a possible discrepancy between OXPHOS metabolic shift in dormant cells and the role of hypoxia in inducing dormancy. We agree with the reviewer that this aspect could be controversial. However, although the dormant phenotype is generally associated with an OXPHOS-dependent metabolic profile, the metabolic program of dormant cells strongly depends on nutrient availability, the tissue of origin, and the microenvironmental cues, leading to the impossibility of defining a unique metabolic setting associated with dormancy for different tumours. We now included an explanation of this apparently contradictory aspect at the end of paragraph 2.2.1 (lanes 663-672 of the “track-changes” file).
- The reviewer asked which known drugs could be tested to target metabolic alterations in dormant tumor cells as an additional option. In order to answer this question, we added a more detailed description of the therapeutic options targeting metabolic adaptations of cancer dormant cells in chapter 3 (lanes 1040-1051 of the “track-changes” file) and we included a new table (Table 2) summarizing the drugs targeting metabolism in dormant cells.
Reviewer 2 Report
Overall, I applaud the authors for such a comprehensive review on an important topic. This is an underappreciated topics that deserves more attention and focus.
Strengths:
- The main strength is in the comprehensive nature.
- The overall organization of sections makes sense.
- Figures are nicely done and easy to understand.
Weaknesses - I recommend addressing the following to make the paper more readable and useful for the reader.
- On page 2, towards the end of the page it appears on line 92 that drug resistance is limited. However, drug resistance may occur from other biologic phenomena, such as increased mRNA and/or protein expression. Additional references here to compare and contrast with dormancy would be valuable.
- Section 1.2 seems to imply that only autophagy or unfolded protein response causes dormancy, but it contrasts with the metabolic descriptions in section 2. A clarification of other potential mechanisms would be helpful.
- In section 1.2, the paragraph is very long and could be better organized. It is recommended to break paragraph up and organize by common gene functions to induce autophagy. This page has a list of genes, but it does not tie together cohesively. Recommend organizing the page by common mechanisms.
- On page 7 in paragraph 2, the focus on T cells seems out of place in this section - a focus on cancer cells seems more appropriate either here or later.
- In section 2, organization of mechanism and comparing/contrasting effects by tumor type would be useful for the readers.
- In section 3, a table listing the potential drugs categorized by mechanism would clarify the potential drugs for the readers.
- In general, a better organization of the sections based on mechanism and/or tumor types would improve the readability of the paper.
- The paper runs a little long. Section 1 could be shortened with a focus more on tumors and a bit less on the microenvironment. While the microenvironment is important, a bit more focus would enhance the readability.
Author Response
Reviewer 2
We thank the reviewer for his/her constructive observations about the manuscript and suggestions that allowed us to ameliorate the review’s structure and the content. We revised the text as suggested and in particular:
- The reviewer appropriately argued that the description of the differences between drug-resistant cells and dormant cells were not exhaustively described in the previous version of the manuscript. We thank the reviewer for this observation. We now included a more detailed description of the mechanism driving drug resistance (lines 114-117 of the “track-changes” file), underling the differences with dormancy. Moreover, we added a discussion of the contribution of dormant cells to the acquisition of drug resistance (lanes 125-132 of the “track-changes” file).
- The reviewer correctly noticed that in Section 1.2 of the previous version of the manuscript, we focused on autophagy and UPR as major mechanisms driving dormancy. However, diverse genetic, genetic and molecular adaptations sustain the dormant phenotype. Therefore, we now included the description of other adaptations driving dormancy at lines 148-195 of the “track-changes” file, mostly focusing on genetic and epigenetic changes.
- We agree with the reviewer that paragraph 1.2 was too long and roughly organized in the previous version of the paper. Therefore, for a better readability of the section, we now divided the section into two paragraphs (intracellular and extracellular mechanisms). Moreover, we relocated the description of the alterations in survival/cell cycle signaling pathways from lines 282-299 to lines 175-188 for a better and more cohesive organization of the text.
- The reviewer correctly commented that the focus on T cells in Section 2.1 was out of place in the section. We agree with the reviewer that the focus on T cells in paragraph 2.1 was too long and out of context. Therefore, we shortened the description of the metabolic profile of dormant T cells in the current version of the paper.
- The reviewer asked for a description of the dormancy-driving mechanism taking account of tumor types. We agree with the reviewer that a more detailed description of factors driving dormancy classified according to tumor type could be helpful through the review. Therefore, we added a table (Table 1) listing the specific markers of CSCs in different tumor types.
- The reviewer suggested adding a table listing the potential drugs categorized by mechanism. We agree with the reviewer that a table listing drugs targeting the metabolic adaptation of dormant cells could clarify the possible therapeutic approaches to the reader. Therefore, we included Table 2 summarizing the drugs targeting metabolism in dormant cells in the current version of the paper.
- The reviewer recommended that a better organization of the sections based on mechanism and/or tumor types improve the paper's readability. We thank the reviewer for underlying this weakness of our text. Therefore, we re-organized section 1.2based on the mechanism driving tumor dormancy to create a more precise description. Moreover, a table listing dormancy-inducing factors in different tumor types may help the reader be geared towards this complex topic.
- The reviewer claimed that the previous version of paper ran a little long. As suggested by the reviewer, we eliminated some examples regarding the implication of the TME in regulation cancer dormancy (paragraph 1.2.) to facilitate the paper's readability
Reviewer 3 Report
The authors has extensively reviewed the metabolic adaptations characterizing the dormant phenotype and supporting tumor re-growth. Further they discussed the role of the metabolic pathways involved in tumor cells dormancy to provide new strategies for selectively targeting these cells to prevent fatal recurrence and maximize therapeutic benefit. T However some minor critiques/suggestions are needs to be addressed
- Do exosomes play any role in cancer dormancy? If possible include few lines about the same
-
Please include a table listing the dormancy factor, cancer types and therapeutic strategies related to metabolic pathway in tumor cells.
Author Response
Reviewer 3
We thank the reviewer for his/her positive observations about the manuscript and suggestions that helped to improve the review’s quality.
- The reviewer asked for a more detailed description of the role of exosomes in driving tumor dormancy. We agree with the reviewer that this is an interesting aspect to be described. Exosomes transfer from the cellular component of the TME to cancer cells plays an important role in dictating the dormant response of cancer cells. Therefore, we added some additional examples underlying the importance of this adaptive mechanism:
- Ono 2014, PMID: 24985346 (lines 171-174 of the “track-changes” file)
- Bliss 2016, PMID: 27569215 (lines 328-331 of the “track-changes” file)
- Walker 2019, PMID: 30683851 (lines 345-350 of the “track-changes” file).
- The reviewer asked to include a table listing the dormancy factor, cancer types and therapeutic strategies related to the metabolic pathways in tumor cells. In order to meet the reviewer request, we included two tables in the current version of the manuscript:
- -Table 1 summarizes the factors characterizing dormant cells according to tumor type/metastatic location.
- -Table 2 summarizes the drugs targeting metabolism in dormant cells.